# Push-Out Method for Micro Measurements of Interfacial Strength in Aluminium Alloy Matrix Composites

**DOI:** 10.3390/ma14175092

**Published:** 2021-09-06

**Authors:** Rafał Kozera, Anna Boczkowska, Zuzanna D. Krawczyk, Paulina Kozera, Maciej Spychalski, Marcin Malek, Robert Kosturek

**Affiliations:** 1Faculty of Materials Science and Engineering, Warsaw University of Technology, ul. Woloska 141, 02-507 Warszawa, Poland; anna.boczkowska@pw.edu.pl (A.B.); zuzanna.krawczyk2.stud@pw.edu.pl (Z.D.K.); paulina.kozera@pw.edu.pl (P.K.); maciej.spychalski@pw.edu.pl (M.S.); 2Technology Partners Foundation, ul. A. Pawinskiego 5A, 02-106 Warszawa, Poland; 3Faculty of Civil Engineering and Geodesy, Military University of Technology, ul. Gen. Sylwestra Kaliskiego 2, 01-476 Warsaw, Poland; marcin.malek@wat.edu.pl; 4Faculty of Mechanical Engineering, Institute of Robots & Machine Design, Military University of Technology, ul. Gen. S. Kaliskiego 2, 00-908 Warsaw, Poland; robert.kosturek@wat.edu.pl

**Keywords:** metal-matrix composites (MMC), fibre/matrix bond, interface/interphase, liquid metal infiltration

## Abstract

The main goal of this work was the evaluation of the interfacial strength of the carbon fibres/aluminium matrix interface dependently on the utilised composite fabrication method, namely high pressure die casting and gas pressure infiltration. In addition, the influence of a Ni-P coating on the C-fibres was investigated. The proposed measurements of the interfacial strength were carried out by means of the “push-out” method. The interfacial strength of the samples fabricated using the high-pressure infiltration method average between 19.03 MPa and 45.34 MPa.

## 1. Introduction

Metal matrix composites (MMCs) reinforced with carbon fibres are superior to traditional metal materials. They can exhibit much better mechanical and thermal properties. Moreover, MMCs are characterised by reduced ability to creeping, especially at elevated temperatures, and good energy absorbance [1]. However, despite many years of research and development focused on aluminium alloy composites reinforced with carbon fibres, the literature still lacks information about their industrial applications. A significant limitation of the practical applications of these composites is the lack of satisfying mechanical properties, resulting from, e.g., harmful reactions on the interface between fibres and matrix. The majority of the publications [2,3,4,5,6] indicate that the phenomena present on the interface are the main factors affecting the composite mechanical properties. Due to the high fibre volume fraction, the fibre-matrix interface significantly influences the MMC’s mechanical properties. Too high intensity of the chemical reaction leads to the formation of aluminium carbide (Al_4_C_3_) precipitations on the interface. They cause a reduced fibre strength, inappropriate stress distribution around the fibres and in consequence improper crack propagation. Therefore, control of the interface strength seems to be crucial.

Applications of the techniques which allow local measurements of the mechanical properties of composites reinforced with fibres by micro and nano intenders are known from the literature. There can be found two main approaches. The first one is based on the direct measurements of Young’s modulus and hardness of the areas around the fibres [7,8], meanwhile, the second one is research focused on the interfacial strength which is carried out using more direct methods, namely fibre push in [9,10], fibre push out [11] and fibre fragmentation [12]. Among them, the fibre push-out test is known to be the most effective investigation method of measuring the interfacial strength between the fibre and the matrix. 

As a result, a few different composite materials such as C/SiC [13] or PET/GF [14] were investigated using micro and nano indentation methods. Moreover, some authors carried out numerical simulations of the debonding strength but mainly on the polymer matrix such as an epoxy/CF [9,15,16,17]. Despite the existence of many publications on the measurement of the interfacial mechanics by means of the abovementioned methods, most of them focused on the non-reactive interfaces with quite a large diameter of fibres, often more than 20 µm. Meanwhile, there is still a wide gap in the interfacial strength measurements where chemical reactions on the interface between the reinforcement and the matrix can occur. Such systems constitute composites with aluminium matrix reinforced by carbon fibres, where in most cases highly intensive chemical reaction between the liquid matrix and the reinforcement is present [18] and thus can significantly affect the interfacial strength. Thus far, only one work dedicated to aluminium alloy matrix composite focused on the interfacial strength measurements was found in the literature [19], however, it was focused on the system reinforced by short fibres. The present work aims to fill this gap by investigating an aluminium alloy composite reinforced by 2D carbon fibre textiles. The composite was fabricated using two different methods, namely high pressure die casting (HPDC) and gas pressure infiltration (GPI) infiltration. Moreover, the influence of different fabrication parameters on the interfacial strength of composites with Ni-P coated and non-coated carbon fibre textiles is evaluated. It should be noted that the HPDC process studied in the present paper was not used previously for the manufacturing of aluminium alloy matrix composites reinforced by carbon fibres. In order to understand the influence of the applied parameters on the interfacial strength, microstructure illustrations of the interfaces are provided. 

## 2. Materials and Methods

### 2.1. Composite Plates

Samples MMC1-3 were fabricated using the HPDC method, while the MMC4 sample was fabricated by means of the GPI method. Each technique of casting method offers the process parameters not available for the second of the compared methods. This gave an opportunity to evaluate the effect of applying extremely different parameters such as time of contact between reinforcement and liquid aluminium alloy (Table 1). In both cases, the same industrial casting aluminium alloy 226D was used to eliminate microstructural differences at the investigated interfaces caused by different chemical compositions of the composite matrix. The most important technological aspect from the carbon fibres point of view in HPDC is a necessity for pre-heating fibres just before casting in order to avoid premature alloy solidification on the surface of the cold reinforcement. Nonetheless, due to the low pressure utilised in the GPI process, it was necessary to use coated fibres to enhance the wettability of the carbon surface by liquid aluminium alloys [20]. Thus, 2D carbon fibres textiles utilised as reinforcement in composite samples MMC2-4 before infiltration were coated with Ni-P coatings deposited in the electroless process. In HPDC, Ni-P coatings are not only improving the wettability but are also considered the protection of the surface of the fibres during the pre-heating process. Detailed procedure of coatings deposition process described elsewhere [21]. The thickness of the Ni-P coatings was approx. 0.5 µm with 2.5 wt% content of phosphorous. Before infiltration by HPDC, the uncoated fibrous reinforcement was pre-heated up to 800 °C, while the coated one was heated to temperatures from 800 up to 1000 °C. The higher pre-heating temperature of the coated fibres was due to their higher thermal conductivity. 

### 2.2. Samples Preparation for Push Out Investigation

Due to the very small diameter of the tested fibres, the scaffold’s shape located under the investigated sample was simplified, as shown in Figure 1. The slots in the scaffold were oblong instead of round as otherwise, it would have been impossible to provide free space exactly under a specific pushed fibre. The appropriate geometry of the slots and distance between them were optimised and verified by means of the Vecco Wyko NT9300 profilometer. The developed solution (Figure 2) provided the right stiffness of the composite plate and simultaneously allowed to localise fibres suitable to push out with free space underneath. Plates with a thickness of 1 mm were cut out of the composites with a wire diamond saw. They were then polished to a thickness between 70 and 170 µm using 2500 grade paper (Table 2). Each polished plate was subjected to the same measuring procedure comprised of thickness measurements using a measuring screw. The evaluation of the results of the push-out tests was performed with SEM SU-70 and HRSTEM S5500 Hitachi with an accelerating voltage up to 30 kV. In order to provide a better observation perspective, all composite plates were tilted about 20°. 

### 2.3. Fabrication of Micro Intender

The fabrication process of the intender was carried out using an Ion Microscope Hitachi FB2100, which ensured a suitable intender shape with nanometer accuracy. The reason for the preparation of a flat intender was better visualisation of the surface of the interface in comparison to other publications in which standard Berkovich intenders were used, as it allowed to push out fibres only for the several dozens of nanometers. Assessment of the interface condition is especially important in chemically reactive systems such as in the case of aluminium alloy composites reinforced by carbon fibres.

Fabrication of the flat intender was carried out in the following stages. In the first one, a standard Berkovich pyramid shape intender was flattened at the top side (Figure 3a). Then, the cutting process of the oval shape with a diameter higher than desired was started (Figure 3b). Subsequently, the material around the intender was sputtered in order to obtain the preferred final height. In the last step, the desired diameter of the intender, i.e., approx. 5.4 µm was cut (Figure 3c,d). The diameter height was precisely adjusted to prevent buckling during indentation tests, as that would cause the failure of the intender. 

### 2.4. Push-Out Test Procedure

Due to the small diameter of the fibres (7 µm), the research required special precision in the samples prepared as well as during the tests. The tests were conducted using a fabricated microintender and realised by means of a Hysitron Ti950 (Bruker Company, Tucson, AZ, USA) equipped with the load head which allows application of the loads up to 7 N. Before the tests, calibration procedure of the new microintender on the silica plate was carried out. Calibration up to 1 N is allowed to measure the stiffness of the microintender and verify if the intender would not fail during the tests with lower loadings. During the investigation, the maximal load which the intender may reach was set on 170 mN. In the next step, tests were carried out on the composite plate but in a place without fibre. As a result, the final confirmation of the usability of the fabricated microintender was achieved. The indentation print observed in Figure 4a exhibited a regular shape, and the load–depth curve was appropriate (Figure 4b). Fracture strength between fibre and matrix was calculated using the Formula (1):(1)$$\tau =\frac{P}{2\pi rt},$$
where: *τ*—fracture strength, *t*—thickness of the composite sample, *P*—load, *r*—radius of the fibre [19]. 

Measurements were carried out on 5 fibres separated from each other by at least 3 diameters of fibre.

## 3. Results

The values obtained in the result of the indentation tests were collected in Table 2. It is obvious that samples fabricated using high pressure die casting method (MMC1–3) exhibit significantly lower interfacial strength (19–40 MPa) comparing to sample MMC4 which was fabricated using the gas pressure infiltration method (>110 MPa). Analysing the values of the loads needed to push out the fibre from the composite samples fabricated by the HPDC method, it can be found that sample MMC1 exhibits the highest interfacial strength among all of the investigated samples. Because no push-out of the fibres from the plate fabricated by the GPI process was found (what is showed later in this section), it was assumed that the interfacial strength is higher than 110 MPa, which corresponds to the maximal load recorded during the tests of this sample.

### 3.1. High Pressure Die Casting, Samples MMC1–MMC3

After each test observations by means of high-resolution scanning electron microscopy of the top and bottom surface of the plates were carried out. In the result, the surface of the MMC1 sample is presented (Figure 5a–c). Push out of fibre can be easily identified. Characteristic grooves at the interface between the fibres and the matrix originating from the carbon fibre surface are visible. Observation of the bottom side of the composite plate (Figure 6d) confirmed that fibre is pushed out through the whole thickness of the composite sample, which validated the results of the tests. Figure 5d presents a typical MMC1 composite plate, load–displacement curve obtained during push-out tests. Brake of the curve means the push-out of the fibre and a failure of the interface, which enabled us to find the maximum loads needed to debond fibres from the matrix at the interface. In the described case the discontinuity of the curve was observed at approx. 100 mN what gave c.a. 40 MPa of interfacial strength. The abrupt increase in the loading force after the break of the load–displacement curves was caused by the end of prepared indenter range and contact of the composite plate with a larger area of the bottom part of indenter. This phenomenon is also typical for the rest of the carried out tests, except the sample MMC4 where no push out of the fibres was observed.

The load–displacement curves registered for MMC1 as well as for MMC3 presented later in this paper exhibit characteristic contraflexure suggesting a break of the interface followed by a push out of fibre. The absence of the contraflexure in MMC2 curves (Figure 7) indicates a very rapid break at the interface. The break of the curve was observed at c.a. 50 MPa which gave approx. 19 MPa (half of the MMC1 interfacial strength).

The MMC2 investigation of the top as well as of the bottom surface of the plate (Figure 6) indicated the same effect of push out as in the case of MMC1. However, in this case, no characteristic grooves at the interface between fibre and matrix were noticed. Alternately, a bright circle inside the fibres was observed. The interface was irregular, non-uniform and significantly different in comparison to Figure 5. 

Similarly to the MMC1 sample, in the case of the MMC3 sample, SEM investigation evidenced grooves similar to the typical surface of the carbon fibres on the surface of the interface. Moreover, some of the debonded pieces of the interface were visible (Figure 7a,b). The bottom side of the sample (Figure 8a–c) confirmed again properly pushed out fibre from the matrix. The break of the interface was also evidenced by a clearly visible crack around the fibre. In Figure 8d, contraflexure was found at approx. 140 mN (45 MPa). On the surface around each fibre, bright circles are visible. They origin from the Ni-P coatings deposited on the surface of fibres before the infiltration process. This conclusion would be confirmed further in this paper.

### 3.2. Gas Pressure Infiltration, Sample MMC4

The SEM investigation of the top surface of the sample confirmed no push-out of the fibre from the matrix as well as indicated a large crack running through the area of the fibre visible in Figure 9a. Hence, contraflexure of the curve was observed at approx. 140 mN for the MMC4 sample which suggested the failure of the interface in the previously tested samples, in this case, indicates the formation of the crack inside the fibre. In the case of sample MMC4 which was fabricated using the gas pressure infiltration method, the course of the load–depth curve looked differently (Figure 9b). No characteristic discontinuity of the curve found in the previous tests was observed. The curve exhibits a shape of a hysteresis loop, indicating no push-out effect of the fibre from the matrix, despite the intender reaching its maximal load value, i.e., 170 mN (>110 MPa). It should be mentioned, that the same results were obtained for all of the investigated fibres of the MMC4 sample.

## 4. Discussion

Values of the interfacial strength measured for MMC1 remain in the range of the interfacial strength of the interface recorded for, e.g., PET reinforced with glass fibres (57.8 ± 6.4 MPa) [14], but they are significantly lower compared to the system of, e.g., aluminium alloy reinforced with Al_2_O_3_ fibres (77 ± 7 MPa). Values registered for sample MMC2 are slightly above the values found for the epoxy/carbon fibre system (13.7 MPa) [15]. The MMC4 values are similar to the aluminium alloy matrix reinforced with SiC fibres with an interface reaction of ca. 100 nm thickness (131 ± 18 MPa) [22,23]. This suggests that the interfacial strength in MMC4 can be much higher than 110 MPa, due to a significantly larger interaction zone compared to the 100 nm indicated in the case of a SiC/Al system.

In order to find an explanation of the observed relationship between the interfacial strength recorded for the different investigated samples, a TEM investigation of the interface between fibre and matrix was performed. A strong dependence of the interface microstructure and data recorded during push out tests is clearly visible. On the MMC1 interface a very thin layer of approx. 40 nm of the chemical reaction between the reinforcement and the matrix is observed. The reinforcement in this sample was uncoated, nevertheless, two different layers may be distinguished. The first one is on the side of the matrix with a thickness of a few nanometers. Its chemical composition was not analysed in the frame of this paper, however, it is suggested that it could be a residue of the epoxy-based sizing, which is commonly deposited on the surface of carbon fibres to protect them, e.g., during transportation and storage [24]. The second one, a thicker layer exhibit an increased content of oxygen (Figure 10) which is the result of an oxidation process during the pre-heating process. The different contrast can be also a result of an oval shape of the fibre. Due to a thickness of a few dozens of nanometers of the sample prepared for TEM and a groovy surface of the fibres, the layer can be observed as a result of the changing thickness of the fibre and the interface. Additionally, an enhancement of the interfacial strength can be caused by a difference in the thermal expansion coefficient of the carbon fibres (radial 12 × 10^−6^/K) and aluminium alloy matrix c.a. 25 × 10^−6^/K). As a result, fibres besides the chemical reaction can be mechanically locked in the matrix due to the matrix shrinkage during cooling.

The interface of MMC2 is presented in Figure 11. As it can be seen, the microstructure of the MMC2 sample differs significantly comparing to the MMC1 interface microstructure (Figure 10). Between the carbon fibre and the aluminium alloy matrix, a reaction zone-wide for approx. 1 µm is present. The TEM investigation evidenced that the structure is porous and discontinuous, similar to a sponge, which is indicated as white areas in Figure 11. Moreover, in the middle, there is located a layer that is wide for approx. 200 nm and appears as a black line. An EDS analysis presented in Figure 12 evidenced that the discontinuous zone is composed of carbon that indicates the structure of carbon fibre, while the black line is composed of nickel and phosphorous that suggest the presence of nickel phosphorous coating was deposited before composite fabrication. The literature review provided an explanation of such a result. The reason is the catalytic graphitisation process which can proceed in the presence of nickel in certain temperatures, usually over 900 °C [25]. As a result of the graphitisation process and sponge-like interaction zone, a significant drop in the interfacial strength value by approx. 50% to 19 MPa may be observed.

In the case of the MMC3 microstructure of the interface between the fibre and the matrix, a continuous layer of the Al–Ni intermetallic phase appeared as a light grey zone (Figure 13). The presence of this phase is reflected in an enhancement of the interfacial strength observed during the push-out tests, despite some voids being present on the side of the matrix.

Inspection of the interface of sample MMC4 explains why no push out of fibres was observed. Figure 14 indicates a significantly developed reaction zone between the fibre and the matrix. Characteristic needle-like precipitations of Al_4_C_3_ were observed, which are also described elsewhere in the literature [26,27] for aluminium matrix composites reinforced by carbon fibres. The presence of the aluminium carbides prevents the sliding of the fibres. Thus, the push-out tests failed and instead of push-out effect, cracks in the range of fibres were found (Figure 9). It is worth noticing that the aluminium carbides are not only growing into the matrix (Figure 14a), but also into the carbon fibre. The dashed line, which is visible in Figure 14b, shows approximately the start of the carbon fibre surface. Despite the application by Ureña et. al [19] a different method for the fabrication of composites based on the small, specifically prepared in the laboratory samples, in this case, as well as in the case of our semi-industrial GPI method, but differently from the HPDC, the lack of both sliding and the push-out effect was found. Comparing the nickel-coated reinforced composites fabricated by Ureña et. al and in this paper in the HPDC process, the influence of the fabrication parameters seems to be less important due to the presence of nickel coating, its intermetallic phases around the fibres and the limitation of aluminium carbides formation.

## 5. Conclusions

Interfacial strength differs depending on the utilised fabrication method as well as on the process parameters. Samples fabricated using high pressure die casting method exhibited significantly lower interfacial strength compared to the composite sample obtained by gas pressure infiltration (MMC4). On the other hand, sample fabricated using gas pressure infiltration exhibits a significantly more developed interface. This prevents the push-out effect and as a result, precludes the proper mechanic behaviour of the composites reinforced by fibres. Instead of the push-out, cracks inside the fibres appear. This limits the sliding of the fibres and explains the considerably low mechanical properties of fabricated composites found by other authors utilising GPI. Based on the obtained results it can be concluded that the microstructure of the interface plays a crucial role in the interfacial strength values. Experimental investigation of the push-out proved that the control of the aluminium carbides formation can impact the sliding of fibres in the matrix since the load is utilised. Very short contact time between the liquid aluminium alloy and the carbon fibres reinforcement presented in the HPDC method can efficiently prohibit chemical reaction between the fibres and the matrix. On the other hand, the presence of the catalytic graphitisation found during the preheating process of the MMC2 reinforcement and the sponge-like microstructure significantly reduce the interfacial strength, thus it should be taken into consideration during the preheating temperature adjustments. The interfacial strength recorded for MMC1 and MMC3 show the high potential of the HPDC method for the future development of the aluminium matrix composites reinforced by the carbon fibres textiles with satisfying mechanical properties so essential for the industry. Since the interfacial strength values obtained for these samples seem to be more promising comparing to, e.g., the GPI method, the HPDC technique should be further developed and optimised.

## Figures and Tables

**Figure 1 materials-14-05092-f001:**
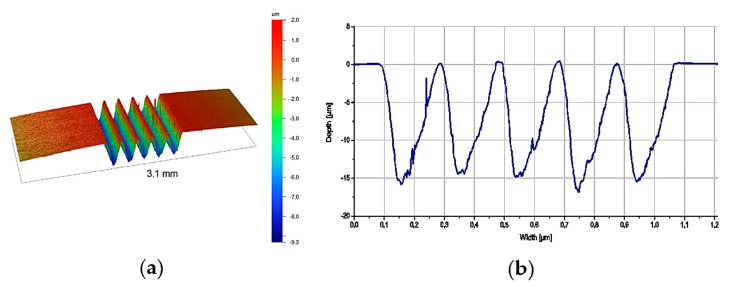
Images of the scaffold: (**a**) general view; (**b**) cross-section with indicated grooves geometry obtained by means of profilometer.

**Figure 2 materials-14-05092-f002:**
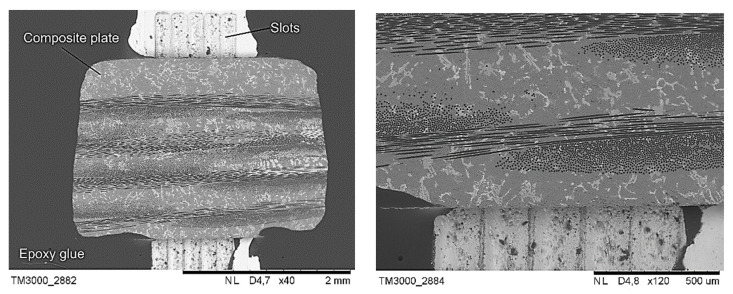
Images of a composite plate stitched to scaffold (light grey) before tests.

**Figure 3 materials-14-05092-f003:**
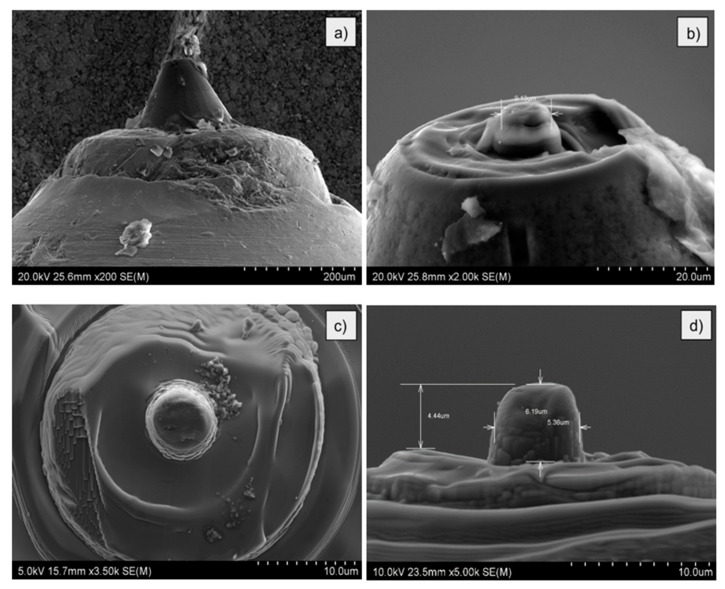
Images of the intender processing stages: (**a**) general view; (**b**) start of cutting process; (**c**,**d**) final result with indicated intender geometry.

**Figure 4 materials-14-05092-f004:**
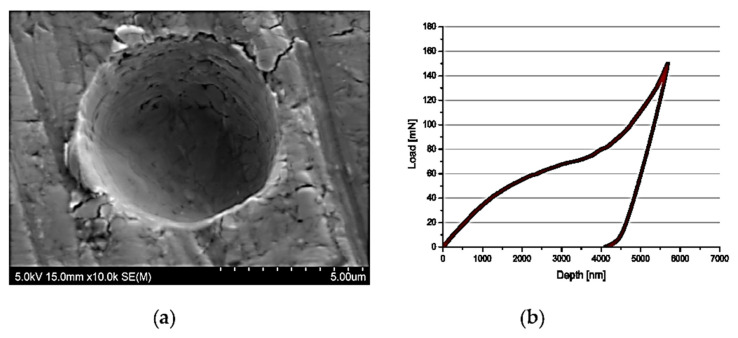
Image of: (**a**) the surface of the sample after preliminary push out tests (**b**) typical curve load–displacement.

**Figure 5 materials-14-05092-f005:**
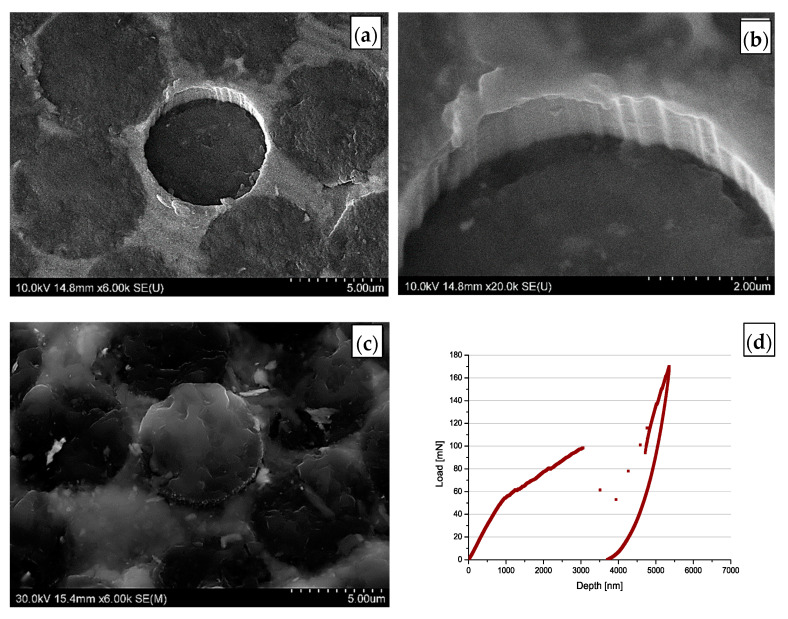
Image of the MMC1 after push-out tests: (**a**,**b**) top; (**c**) bottom surface of plate; (**d**) typical curve obtained during push out tests with MMC1 sample.

**Figure 6 materials-14-05092-f006:**
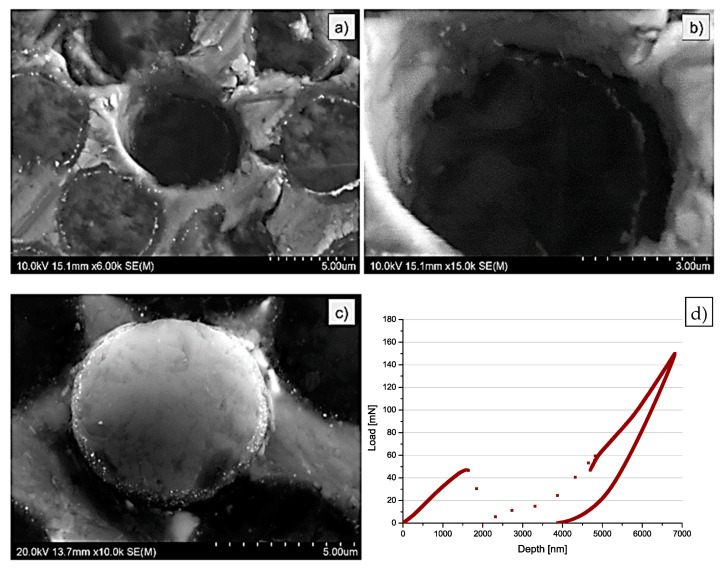
SEM images of the MMC2 sample after push-out tests: (**a**,**b**) top; (**c**) bottom surface; (**d**) typical curve obtained during push out tests with MMC2 sample.

**Figure 7 materials-14-05092-f007:**
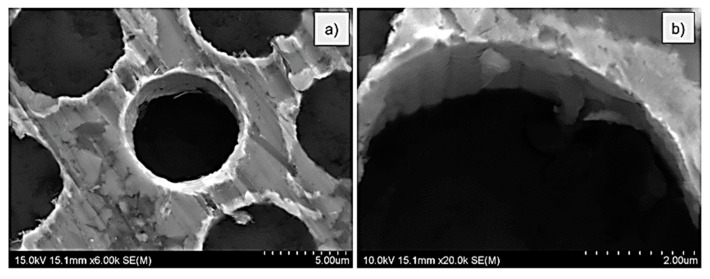
Top surface SEM images of the MMC3 sample after push-out tests: (**a**) BSE mode; (**b**) SE mode.

**Figure 8 materials-14-05092-f008:**
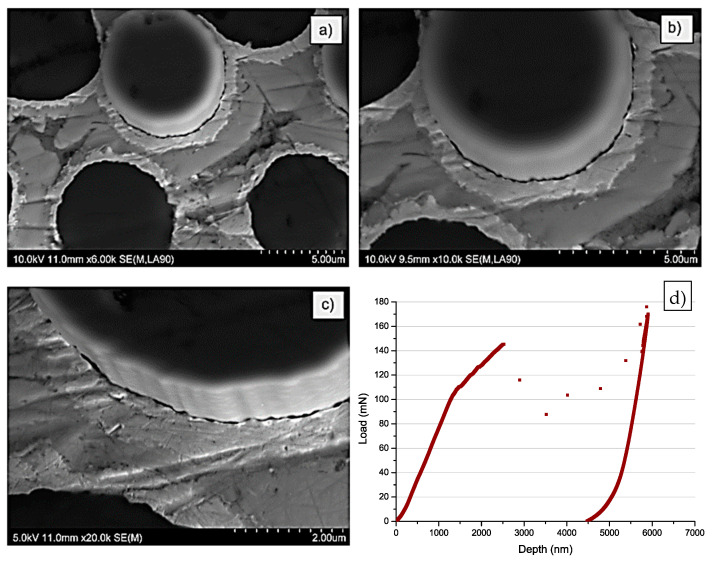
Bottom surface SEM images of the MMC3 sample after push-out tests: (**a**,**b**) BSE mode; (**c**) SE mode; (**d**) typical curve obtained during push out tests with MMC3 sample.

**Figure 9 materials-14-05092-f009:**
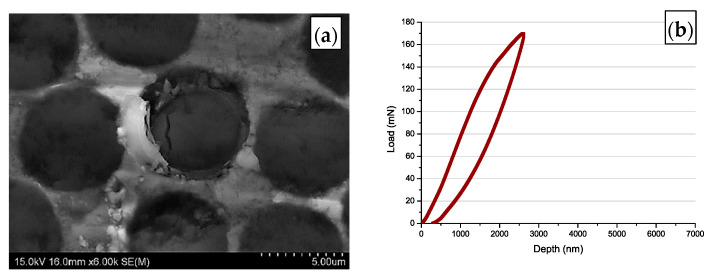
Typical curve obtained during push out tests with MMC4 sample (**a**), top surface SEM images of the MMC4 sample after push-out tests (**b**).

**Figure 10 materials-14-05092-f010:**
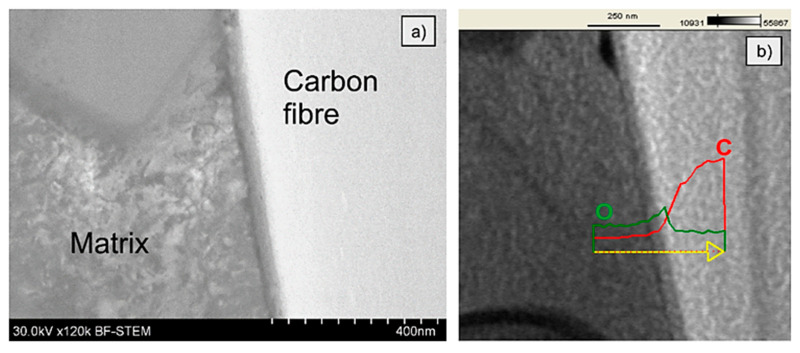
Bright-field TEM image of the interface between carbon fibre and aluminium alloy matrix of MMC1 sample: (**a**) before the push-out test; (**b**) linear EDS analysis.

**Figure 11 materials-14-05092-f011:**
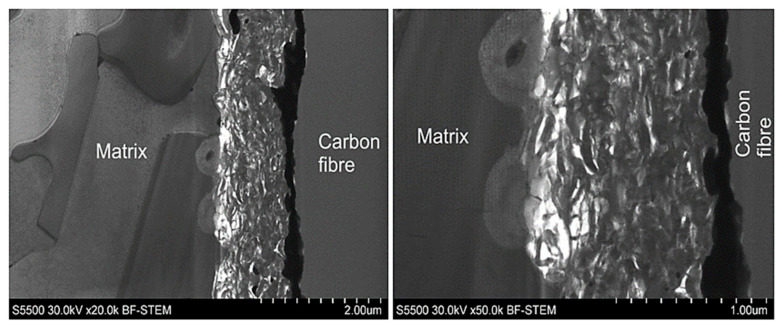
TEM images of the interface between fibre and aluminium alloy matrix of MMC2 sample before the push-out test.

**Figure 12 materials-14-05092-f012:**
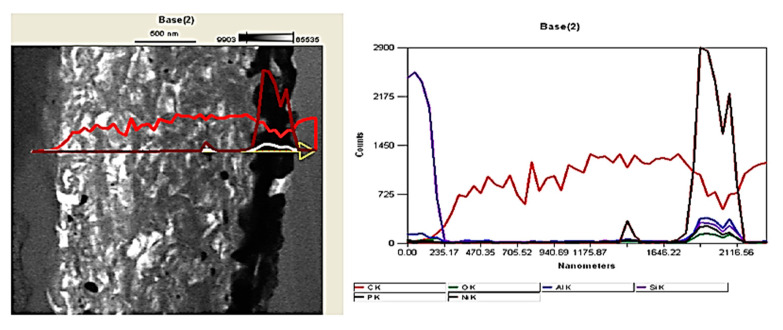
EDS linear analysis of the MMC2’s fibre-matrix interface.

**Figure 13 materials-14-05092-f013:**
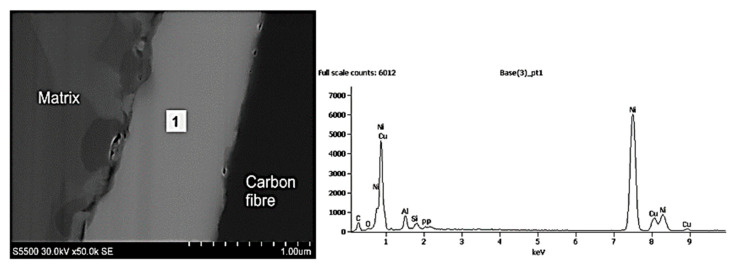
EDS analysis of the interface between the fibre and the aluminium alloy matrix of MMC3 sample before push-out test.

**Figure 14 materials-14-05092-f014:**
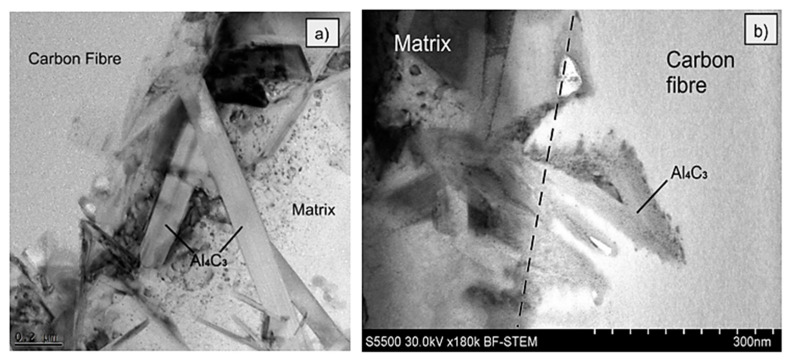
TEM images of the interface between the fibre and the aluminium alloy matrix of MMC4 sample with indicated aluminium carbides penetrating: (**a**) matrix; (**b**) carbon fibre.

**Table 1 materials-14-05092-t001:** Fabrication parameters of the investigated samples.

Sample	Carbon Fibres	Pre-Heating Temperature (°C)	Infiltration Temperature (°C)	Infiltration Pressure (MPa)	Time of Contact (s)
MMC1	Uncoated	800	700	60	<1
MMC2	Ni-P coated	1000	700	60	<1
MMC3	Ni-P coated	800	700	60	<1
MMC4	Ni-P coated	720	720	15	1800

**Table 2 materials-14-05092-t002:** Results of push-out tests for samples fabricated by high pressure die casting and gas pressure infiltration methods.

Sample	Sample Thickness (μm)	Load (mN)	Interfacial Strength τ (MPa)
MMC1	173 ± 2	110.43 ± 19.4	40.13 ± 6.9
MMC2	129 ± 3	53.95 ± 11.4	19.03 ± 4.0
MMC3	140 ± 2	139.54 ± 20.3	45.34 ± 6.6
MMC4	70 ± 1	>170	>110.5

## Data Availability

Not applicable.

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
