# Peer review of "Push-Out Method for Micro Measurements of Interfacial Strength in Aluminium Alloy Matrix Composites"

_materials, 2021, doi:10.3390/ma14175092_

Round 1

Reviewer 1 Report

1. Line 102: Check for miswrite. (fibre,s)

2. Load and Interfacial strength τ of MM2 specimen are too small in Table. What is the cause? Comparing with Table 1, it was judged that it was probably the difference in pre-heating temperature, and was the difference in pre-heating temperature the main factor?

3. Load-displacement curves of MMC 1,2 and 3 seem to be incomplete. The cause is written in the treatise. But is this result credible? Can you add references?

4. Line 177: [approx.] seems to be approximately. But what does [c. a.] stand for? 

5. Is it possible to easily schematize the reinforcement mechanism of MM4? Attach the schematized image to Fig. 18.

Author Response

  1. Line 102: Check for miswrite. (fibre,s)                                                           A: Thank you for comment, Done.
  2. Load and Interfacial strength τ of MM2 specimen are too small in Table. What is the cause? Comparing with Table 1, it was judged that it was probably the difference in pre-heating temperature, and was the difference in pre-heating temperature the main factor?

A: Thank you for this question. Yes the difference in pre-heating temperature which caused observed porosity between matrix and fibre (fig 15) was the main factor affecting lower load and interfacial strength.

  1. Load-displacement curves of MMC 1,2 and 3 seem to be incomplete. The cause is written in the treatise. But is this result credible? Can you add references?

A: Thank you for this questions. Yes they are credible. The discontinuity of the curves of MMC 1,2 and 3 means moment of push out of the fibre. Similar observations and description was described in reference [19]: Ureña, A.; Rams, J.; Escalera, M.D.; Sánchez, M. Characterization of interfacial mechanical properties in carbon fiber/aluminium matrix composites by the nanoindentation technique. Compos. Sci. Technol. 2005, 65, 2025–2038, doi:10.1016/J.COMPSCITECH.2005.04.013.

  1. Line 177: [approx.] seems to be approximately. But what does [c. a.] stand for? 

A: Yes you are right, approx.= approximately, c.a.=circa about

  1. Is it possible to easily schematize the reinforcement mechanism of MM4? Attach the schematized image to Fig. 18.

A: Thank you for this question, we marked schematized line between matrix and fibre to show how developed border is which is responsible for reinforcement mechanism of MMC4, so it is not possible to easier schematize it.

Reviewer 2 Report

This is an interesting study that investigated the interfacial strength in carbon fiber-reinforced Al matrix composites by a push out method. It was found that the consolidation process has great effect on the interfacial structure and consequent interfacial bonding strength. Some revisions are suggested before publication.

1. Based on the load transfer strengthening mechanism, the composite strength improved with the interfacial strength (B. Chen, et al. Composites Science and Technology, 2015, 113, 1-8). Therefore, the reviewer is doubtful about the conclusion that MMC4 with the interfacial Al4C3 would have lower strength than other composites. A more solid conclusion can be drawn by directly comparing the mechanical properties of the four composites in the present study, instead of discussion the scattering data from the literature. The reviewer expects this significant work.

2. The figures need to be arranged to be more tight. For example, Figs. 5, 7, 9 and 12 can be combined, or each of them be put together with the SEM images. Figs. 15 and 16 can be combined.

3. The use of MMC4 in the abstract has to be avoided.

4. The element lines in Fig. 16b need clearer interpretation.

5. Macroscopic interfacial analyses, such as XRD and Raman may be useful to better understand the overall microstructure characteristics.

Author Response

  1. Based on the load transfer strengthening mechanism, the composite strength improved with the interfacial strength (B. Chen, et al. Composites Science and Technology, 2015, 113, 1-8). Therefore, the reviewer is doubtful about the conclusion that MMC4 with the interfacial Al4C3 would have lower strength than other composites. A more solid conclusion can be drawn by directly comparing the mechanical properties of the four composites in the present study, instead of discussion the scattering data from the literature. The reviewer expects this significant work.

A: Thank you for your comment. Of course you are right but it is only true for specified value of strength between fibre and matrix. In the best case, fibres should have possibility to move from the matrix in certain value of strength. In the case of presence of Al4C3 precipitation which are brittle from nature and needle like in shape, transfer of loading between fibres and matrix is interrupted and lead to brittle nature of failure what decrease strength of tested composite. It is very common phenomena in the case of aluminium composites reinforced with carbon fibres described elsewhere e.g. in ref [5]: Lancin, M.; Marhic, C. TEM study of carbon fibre reinforced aluminium matrix composites: Influence of brittle phases and interface on mechanical properties. J. Eur. Ceram. Soc. 2000, 20, 1493–1503, doi:10.1016/S0955-2219(00)00021-2. Ref [6] Vidal-Sétif, M.H.; Lancin, M.; Marhic, C.; Valle, R.; Raviart, J.-L.; Daux, J.-C.; Rabinovitch, M. On the role of brittle interfacial phases on the mechanical properties of carbon fibre reinforced Al-based matrix composites. Mater. Sci. Eng. A 1999, 2, 321–333. Thank you also for the suggestion about mechanical properties of the presented composites. Authors decided to not include those results to keep focus only on the interface between carbon fibres and matrix. Results of mechanical tests and explanation of those findings will be the main subject of the next publication which is almost prepared for review.

  1. The figures need to be arranged to be more tight. For example, Figs. 5, 7, 9 and 12 can be combined, or each of them be put together with the SEM images. Figs. 15 and 16 can be combined.

A: Thank you for your comment. We put together images except figs 15 and 16 to make sure that they will be clearly visible and easy to analyse in the form as they are.

  1. The use of MMC4 in the abstract has to be avoided.

A: Thank you for your comment. Done.

  1. The element lines in Fig. 16b need clearer interpretation.

A: Thank you for your comment. Done.

  1. Macroscopic interfacial analyses, such as XRD and Raman may be useful to better understand the overall microstructure characteristics.

A: Thank you for this advice, but taking into consideration complexity of article we will use those technique in another article which is in preparation and focus more around macroscopic interpretation and combination with mechanical properties of whole composites.